# Vascular and Transpiration Flows Affecting Apricot (*Prunus armeniaca* L.) Fruit Growth



Andrea Giovannini [1], Melissa Venturi [1,*], Saray Gutiérrez-Gordillo [2], Luigi Manfrini [1], Luca Corelli-Grappadelli [1] and Brunella Morandi [1]

1 Department of Agricultural and Food Sciences (DISTAL), University of Bologna, V. le Fanin 46, 40127 Bologna, Italy; andrea.giovannini11@studio.unibo.it (A.G.); luigi.manfrini@unibo.it (L.M.); luca.corelli@unibo.it (L.C.-G.); brunella.morandi@unibo.it (B.M.)
2 Center "Las Torres", Andalusian Institute of Agricultural and Fisheries Research and Training (IFAPA), Carretera Sevilla-Cazalla Km 12.2, Alcalá del Río, 41200 Sevilla, Spain; saray.gutierrez@juntadeandalucia.es
* Correspondence: melissa.venturi@unibo.it

**Abstract:** Fruit growth is a biophysical process that depends mainly on the daily balance between vascular (xylem and phloem) and transpiration flows. This work examines the seasonal and daily behaviour of apricot fruit growth as well as their vascular and transpiration flows. Seasonal patterns of the shoot and fruit growth, as well as fruit surface conductance and dry matter accumulation, were monitored at regular times intervals during the season on "Farbela" and "Ladycot" cultivars. In addition, the daily courses of leaf and stem water potentials and leaf gas exchanges were monitored at 66 and 109 DAFB. On the cultivar "Farbela", the daily patterns of phloem, xylem, and transpiration flow to and from the fruit were determined through precise and continuous monitoring of fruit diameter variations. Branch sap flow was also determined through the thermal balance method. Apricot fruit growth showed a double sigmoid pattern, typical of other drupaceous species. Stem and leaf water potential maintained values above the stress threshold, and in the last part of the season, the leaf photosynthetic rate increased. Leaves received higher sap flow in the morning and at midday, while a higher amount of xylem water was moved to the fruit in the late afternoon. Fruit showed high transpiration rates, which led to fruit shrinkage during the warmest hours of the day. High xylem inflows balanced the transpiratory losses, while phloem import was lower and occurred mainly during the mid-day hours. As a result, the fruit grew mostly in the late afternoon and night, and its growth was sustained mainly by xylem fluxes, which represented over 90% of the fruit's total inflows. Later in the season, fruit transpiration and xylem flow decreased but did not stop even at harvest. Phloem import increased its importance throughout the season and, in the final stages, accounted for 36% and then 66% of the daily relative contribution to fruit growth. This knowledge represents a starting point to improving apricot orchard management in terms of irrigation and fertilisation.

**Keywords:** fruit growth; fruit transpiration; apricot; phloem; xylem





## 1. Introduction

Apricot (*Prunus armeniaca* L.) is one of the most cultivated stone fruits worldwide. Over the last decades, the breeding programs led to the commercialisation of several new varieties with a wider maturation calendar which allows for diversifying the supply during the year [1]. Together with the advancement in the genetic sector, the orchard management has also been improved thanks to new training systems [2] and deficit irrigation strategies, which allowed an increased fruit quality [3,4].

Fruit growth can be considered the result of several biophysical and biochemical processes supporting water and dry matter accumulation in the fruit [5]. Therefore, fruit diameter variation over a definite period of time can be viewed as the net contribution of (i) phloem import, which normally assumes positive values, (ii) xylem flow, whose values could be either positive or negative, and (iii) transpiration flow, characterised by negative

values [5]. The effects of dry matter gain and loss from fruit photosynthesis and respiration on fruit diameter variation can be considered negligible on a daily scale [6].

Phloem and xylem flows are driven by turgor pressure and osmotic concentration gradients from source to sink organs [7–9]. Assimilates synthesised in the leaves are loaded into phloem vessels and carried out to sink organs by symplasmic or apoplasmic mechanisms. The symplasmic phloem unloading strategy passively unloads the assimilates to sink cells either by diffusion or by mass flow, depending on whether their transport responds to solutes concentration or turgor pressure gradients, respectively. In contrast, the apoplasmic phloem unloading brings assimilates to the fruit thanks to specific cell membrane transporters, which require metabolic energy [10–12].

Fruit transpiration is a physical process strictly dependent on fruit surface area, surface conductance, and environmental conditions [13–15]. In general, transpiration causes fruit shrinkage, allowing the fruit to decrease its water potential and can help xylem inflows and the process of phloem symplasmic unloading [13,16].

The balance between incoming and outcoming fluxes leads to fruit swelling or fruit shrinkage, depending on whether the balance is positive or negative, respectively [16]. Since vascular flows depend mostly on environmental factors, fruit diameter variation also changes widely during the day, alternating from periods of growth to periods of shrinkage [5,17,18]. Water and assimilates are translocated to the fruit by xylem and phloem vessels, while fruit epidermis transpiration is the main outgoing flux. Reverse water flow from fruit to leaves is also possible via xylem backflow, and it has been already observed in some fruit species, including citrus [19,20], apple [21], grape [22,23], and kiwifruit [18]. This phenomenon usually occurs during the warmest hours of the day, when leaf and thus stem water potential becomes more negative than the fruit.

In apples, transpiration is greatly reduced near harvest, when the cuticle becomes much thicker and fruit growth is supplied only by phloem flows [13,21]. Similar behaviour occurs in sweet cherry, in which growth is sustained more by the xylem in stage II, while in stage III, the phloem increases its importance while the fruit transpiration reaches values close to zero [24]. In kiwifruit, transpiration progressively decreases during the season because of the suberisation of an outer cell layer of the fruit and biosynthesis of wax [18,25–27]. In peach, the fruit continues to lose water at higher specific rates than apple and kiwifruit [14,28,29] and its growth is sustained by both xylem and phloem flows until harvest [5]. In apricot, fruit transpiration decreases during the season, but, unlike apple and kiwifruit, in the last stage of development, the fruit continues to lose water [30], resembling the peach growth strategy [5,28]. Phloem flow in apricot is supposed to be maintained throughout the season because the accumulation of phloem-mobile nutrients (K and Mg) continues despite the decline of the transpiration rate [30]. Sugar content in the fruit changes during the season; sucrose firstly increases and then decreases, while glucose and fructose concentrations show the opposite behaviour [31]. The main genes for sugar transport and accumulation have been identified, and it has been demonstrated that they work continuously throughout the development and ripening of the fruit but at different intensity rates [31].

Despite the state of the art methods, there is still a lack of information on how xylem, phloem, and transpiration flows account for fruit growth in apricot.

The aim of this study is to elucidate the strategy of apricot fruit growth on a seasonal and a daily scale, determining their water relations, leaf gas exchanges, as well as quantifying xylem, phloem, and transpiration flows to/from the fruit.

## 2. Materials and Methods

### 2.1. Plant Material and Environment Data

The study was conducted during the 2021 vegetative growing season at the experimental farm of the University of Bologna, located in Cadriano, Bologna, Italy (44°32′53″ N 11°24′45″ E).

The trial was set at a four-year-old apricot orchard (*Prunus armeniaca* L.). A total of 12 plants were selected, six of cv. Farbela and six of cv. Ladycot, both grafted on Myrobalan 29C (*Prunus cerasifera* Ehrh.). The orchard was covered with an antihail net. Trees were trained at an open vase, with a density of 1000 trees ha$^{-1}$ (5 m × 2 m). Standard cultural practices were applied for pruning and fertilisation; thinning was not performed since spring frost reduced the initial crop load by about 50%. Irrigation was scheduled based on the suggestions provided by the regional software platform "Irrinet" (Consorzio per il Canale Emiliano Romagnolo—CER). Temperature, relative humidity, and rainfall data were collected from a weather station (A840 Base Station, Adcon Telemetry GmbH, Klosterneuburg, Austria) placed 50 m far from the orchard. With the collected data, vapour pressure deficit (VPD) was calculated using the following equation (Equation (1)):

$$VPD = e_s \left( \frac{RH}{100} \right)$$
(1)

where $e_s$ is the saturation vapour pressure, and $RH$ is the relative humidity.

Full bloom occurred on 12 March in both cultivars, while the fruit was harvested on 30 June, 110 days after full bloom (DAFB) in Ladycot and on 21 July, 131 DAFB in Farbela.

### 2.2. Fruit and Shoot Seasonal Growth

The growth of 96 fruit (eight per tree) and 48 shoots (four per tree), randomly selected on the two sides of the canopy, was monitored on a weekly basis using a digital calliper and a measuring tape, respectively.

For each fruit, diameter (D) data were converted to weight (*W*) by the following conversion equation (Equation (2)):

$$W(g) = a \cdot D \ (mm)\textasciicircum b$$
(2)

where *a* and *b* were 0.0006 and 2.9946, for cv. Farbela, and 0.0017 and 2.7601 for cv. Ladycot, respectively. These equations were obtained by regressing diameter and weight data from fruit collected periodically during the whole growing season. The $R^2$ of the relationship was 0.95 and 0.97, respectively.

Fruit and shoot absolute growth rates (AGR) (g fruit$^{-1}$) were calculated at each recording time (*t*) using the following equation (Equation (3)):

$$AGR_{t1} = \frac{FW_{t1} - FW_{t0}}{t_1 - t_0}$$
(3)

were $FW_{t1}$ was the fruit weight at recording time $t_1$, and $FW_{t0}$ was the fruit weight at the previous recording time.

### 2.3. Water Relations

Stem ($\Psi_{stem}$) and leaf ($\Psi_{leaf}$) water potentials were monitored at 59, 66, 84, 109, 112, 115, 117, 119, 122, 124, 129, and 131 DAFB at solar midday (13:00 h), on at least four trees for each of the two cultivars, using a Scholander (Soilmosture Equipment Corp., Santa Barbara, CA, USA) pressure chamber. At 66 and 109 DAFB, full-day measurements were also performed, with data collected at 04:00, 9:30, 12:00, and 15:00 h.

$\Psi_{leaf}$ were measured on one well-exposed shoot leaf per tree following the methodology proposed by Turner and Long [32]. $\Psi_{stem}$ were measured on the same trees; one leaf per tree placed in the inner part of the canopy, very close to the main stem, was chosen and covered with aluminium foil at least 90 min before the measurement to allow equilibration with the stem, according to the methodology described by McCutchan and Shackel [33] and by Naor et al. [34].

### 2.4. Leaf Gas Exchanges

Leaf gas exchanges were measured in concurrence with the water potential measurements using an open-circuit leaf gas-exchange system equipped with a LED light source (Li-COR 6400, LI-COR, Lincoln, NE, USA). Measurements were carried out on the same plants monitored with the water potential, choosing one fully illuminated and developed leaf. During each measurement, light intensity was maintained constant at the natural photosynthetically active radiation (PAR) level experienced by the leaves immediately before the measurements, and the $CO_2$ concentration was set at 400 mg kg$^{-1}$. The main parameters considered in the data elaboration were net photosynthesis, transpiration, and stomatal conductance ($g_s$).

### 2.5. Xylem, Phloem and Transpiration Flows

Fruit vascular and transpiration flows were determined in cv. Farbela, in 3 different periods: 30 April–9 May (49–58 DAFB), 28 May–7 June (77–87 DAFB), and 28 June–21 July (108–131 DAFB). In each period, diameter variations over time were simultaneously monitored on 12 well-exposed fruit placed on 4 trees, on both sides of the canopy, following the methodology proposed by Lang [21]. On each tree, there was a fruit (three fruits in total) subjected to one of the following conditions: "intact" (with normal vascular connections), "girdled" (with the phloem connection severed), and "detached" (with all vascular connections severed). After a maximum of two days, fruits were substituted, and three more fruit were subjected to the same conditions. The values obtained from the three fruits were then used to calculate xylem and phloem flows. Fruit diameter variations over time were monitored at 15 min intervals by using custom-built gauges interfaced with a wireless data-logger system (Winet srl, Cesena, Italy). The gauges consisted of a light, stainless steel frame supporting a variable linear resistance transducer (Megatron Elektronik AG & Co., Munich, Germany). Temperature effects on the frame and the sensor were tested and showed negligible errors under normal field conditions [35].

This method is based on the further assumptions that xylem flow is not affected by girdling, and transpiration rate is not affected by detachment. Fishman et al. [14] report how the first assumption can lead to some systematic errors, causing under- and overestimation of phloem and xylem flows, respectively. However, these errors seem to be limited to specific times during the day and, to date, this is the only method that allows estimation of vascular and transpiration flows in a field-short times scale and on a statistically sound number of samples.

The relative changes of fruit fresh weight in each time interval ($t$) were then calculated for each fruit: normal ($N$), girdled ($G$) and detached ($D$); xylem ($X$), phloem ($P$), and transpiration ($T$) flows were computed using the following equations (Equations (4)–(6)):

$$P_t = N_t - G_t \tag{4}$$

$$X_t = G_t - D_t \tag{5}$$

$$T_t = D_t \tag{6}$$

Fruit growth rate, phloem, xylem, and transpiration flows were expressed both as weight changes per whole fruit (g fruit$^{-1}$) and per unit of fruit weight (g g$^{-1}$).

### 2.6. Sap Flow Measurements

Sap flow was determined through the thermal balance method described by Sakuratani [36]. Measurements were carried out in concurrence with fruit gauges measurement to compare the daily amount and patterns of the xylem sap flowing in the whole branch to the xylem flowing to the fruit (determined through the methodology described above).

In each period, four sap flow sensors [37] were placed on 10 to 19 mm diameter branches carrying at least one fruit on the same trees where the fruit gauges were. Data were recorded by a CR10X data logger (Campbell Scientific Inc., Logan, UT, USA) each

minute and automatically averaged every 15 min, from 3:30 h to 22:30 h. During each monitoring period, the system worked continuously. Sap flow was then normalised per unit leaf area and expressed as g m$^{-2}$ h$^{-1}$.

The leaf area of each monitored branch was determined by calculating the area of six randomly picked leaves using ImageJ software (https://imagej.nih.gov/ij), then multiplying the average of the leaf area by the total number of leaves counted on the branch where the sap flow sensor was installed.

### 2.7. Fruit Surface Conductance and Dry Matter Content

Ten fruits for each of the two cultivars were sampled every week throughout the growing season. After mass and diameter measurement, the fruits were placed in a room whose temperature and relative humidity were monitored by a data logger (Lascar EL-USB-1, Lascar Electronics Ltd., Whiteparish, UK), and after 24 h, the fruits were weighed again. Surface conductance ($g_c$) was calculated following Fishman and Génard's [16] equation (Equation (7)):

$$g_c = \frac{T_f\, R\, T}{A_f\, M_w\, VPD} \tag{7}$$

where $T_f$ is the rate of water loss per unit of time as a result of transpiration, $R$ is the gas constant, $T$ is the absolute temperature (K), $A_f$ is the fruit area (m$^2$), $M_w$ is the molecular mass of water (18 g mol$^{-1}$), and VPD is the vapour pressure deficit (kPa).

Fruit area ($A_f$) was calculated assuming the fruit shape was similar to an ellipsoid with three different axes and computing the area using the equation (Equation (8)):

$$A_f = \left( \frac{r_1^p\, r_2^p + r_1^p r_3^p + r_2^p r_3^p}{3} \right)^{1/p} \tag{8}$$

where $r^1$, $r^2$ and $r^3$ are the three semiaxes of the ellipsoid, and $p$ is a constant value of 1.6075.

The same fruits, after surface conductance measurement, were used to determine flesh dry matter content by slicing and drying them at 65°C until constant weight in a fan-forced oven. Dry matter content (DMC %) was then calculated for each sampling date as (Equation (9)):

$$DMC(\%) = \frac{dry\ mass}{fresh\ mass} 100 \tag{9}$$

### 2.8. Statistical Analysis

All data considered were analysed with a completely random design, and averages and standard error (SE) were calculated per each day of measurement. For each date of measurement, cultivars were compared through a Student's *t*-test.

## 3. Results

### 3.1. Fruit and Shoot Seasonal Growth

Apricot fruit growth showed a double-sigmoid pattern (Figure 1a), more pronounced in Farbela than in Ladycot because of its longer fruit growing season. Both cultivars grew fast until 60 days after full bloom (DAFB). Ladycot had a slight slowdown in growth between 61 and 75 DAFB, then rapidly switched to cell expansion (Figure 1a). Farbela had a growth slowdown between 61 and 88 DAFB and then increased its absolute growth rate (AGR) up to 3.3 g d$^{-1}$ (Figure 1b). The highest AGR values were recorded in the last stages of fruit growth for both cultivars (Figure 1b).

Seasonal shoot growth showed similar behaviour to fruit growth (Figure 2a). Higher AGR values were recorded at the beginning and in the last part of the season. In the period between 70 and 100 DAFB, both cultivars showed a decreasing shoot AGR, with significantly lower values in the cultivar Farbela. For both cultivars, shoot AGR increased

again, reaching maximum values of about 0.7 cm day$^{-1}$ between 110 and 120 DAFB, then it decreased again at the end of the season (Figure 2b).

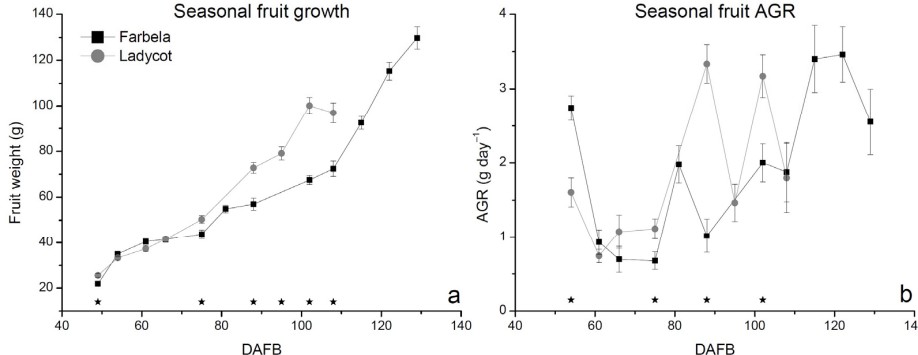

**Figure 1.** Seasonal fruit growth (+SE) (**a**) and fruit absolute growth rate (AGR) (**b**) of Farbela (squares) and Ladycot (circles). Diameter values were converted to mass by Equation (2). Each point represents the mean of 48 fruit. SE bars smaller than symbols are not visible. Stars represent statistical differences for Student's *t*-test $p < 0.05$.

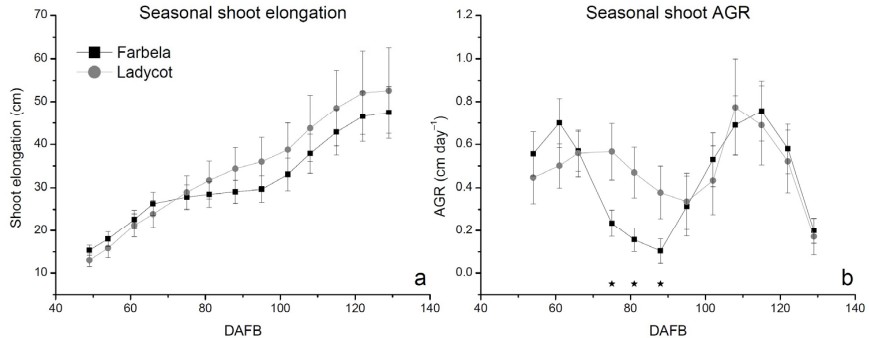

**Figure 2.** Seasonal shoot growth (+ SE) (**a**) and shoot absolute growth rate (AGR) (**b**) of Farbela (squares) and Ladycot (circles). Each point represents the mean of 24 shoots. Stars represent statistical differences for Student's *t*-test $p < 0.05$.

*3.2. Water Relations*

Stem and leaf water potentials tended to decrease throughout the season (Figure 3). On a daily scale, at 66 DAFB, $\Psi_{leaf}$ decreased during the day from predawn until 15:00 h, while $\Psi_{stem}$ decreased only at mid-day, reaching −0.6 and −0.5 MPa for Farbela and Ladycot, respectively (Figure 4a).

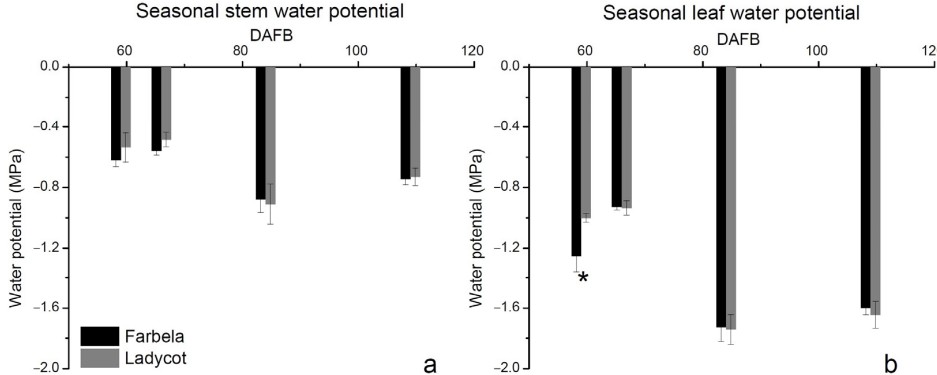

**Figure 3.** Seasonal stem (**a**) and leaf (**b**) water potentials (+SE) of Farbela (black bar) and Ladycot (grey bar), measured at 59, 66, 84, and 109 DAFB. Each point represents the mean of 4–6 replicates. Stars represent statistical differences for Student's *t*-test $p < 0.05$.

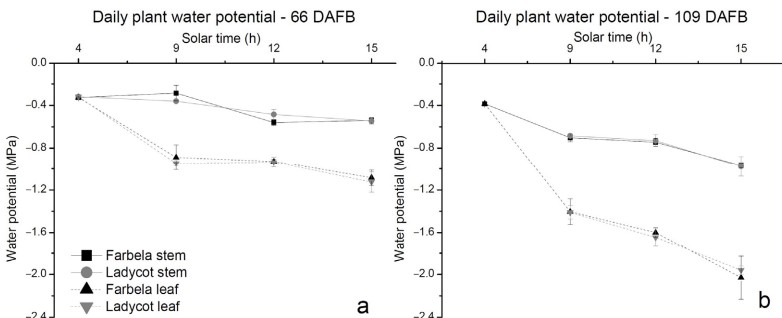

**Figure 4.** Daily stem (squares for Farbela and circles for Ladycot) and leaf (triangles) water potentials (+SE), measured at 66 DAFB (**a**) and 109 DAFB (**b**). Each point represents the mean of 4–6 replicates. SE bars smaller than symbols are not visible.

$\Psi_{leaf}$ was always more negative than the stem. At 109 DAFB, the stem-to-leaf water potential gradient increased during the day from 0.7 MPa at 9:00 h to 1.1 MPa at 15:00 h on Farbela (data not shown). However, at 66 DAFB stem-to-leaf water potential gradients were higher in the morning and in the afternoon and lower at 12:00 h in both cultivars (data not shown).

Water potentials were not affected by the cultivar, showing similar values on a daily and on a seasonal time scale, with no statistical differences, except for $\Psi_{leaf}$ at 59 DAFB (Figure 4b).

### 3.3. Leaf Gas Exchanges

During the whole season, leaf photosynthesis (Figure 5a), transpiration (Figure 5b), and stomatal conductance (Figure 5c) increased in both cultivars. Ladycot showed generally higher values in all leaf gas-exchange parameters during all seasons, although environmental and management conditions were the same for both cultivars. Ladycot and Farbela leaf photosynthetic rates at 59 DAFB were 18.92 and 14.9 $\mu$mol m$^{-2}$ s$^{-1}$, respectively, while later in the season, at 109 DAFB, they increased to 28.7 and 25.0 $\mu$mol m$^{-2}$ s$^{-1}$, respectively (Figure 5a). Transpiration and stomatal conductance increased as a consequence: transpiration rose from 2.4 and 2.1 to 10.9 and 10.1 mmol m$^{-2}$ s$^{-1}$ for Ladycot and Farbela, respectively (Figure 5b), while stomatal conductance increased in both cultivars from about 0.1 to 0.4 mol m$^{-2}$ s$^{-1}$ (Figure 5c).

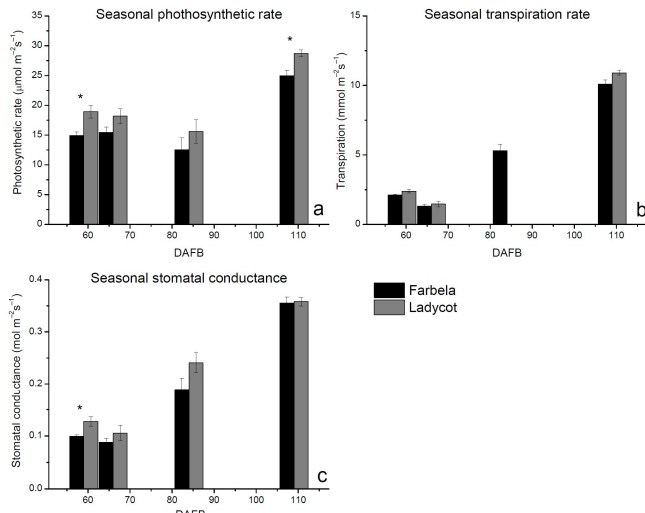

**Figure 5.** Seasonal leaf net photosynthesis (+SE) (**a**), leaf transpiration (+SE) (**b**) and stomatal conductance (g$_s$) (+SE) (**c**) of Farbela (black bar) and Ladycot (grey bar), leaves measured at 59, 66, 84 and 109 DAFB. Each point represents the mean of 4–6 replicates. Stars represent statistical differences for Student's *t* test $p < 0.05$.

On a daily scale, leaf photosynthetic rate reached maximum values around mid-day (Figure 6) while, at 109 DAFB, it did not show a drastic drop even at 15:00 h (Figure 6b).

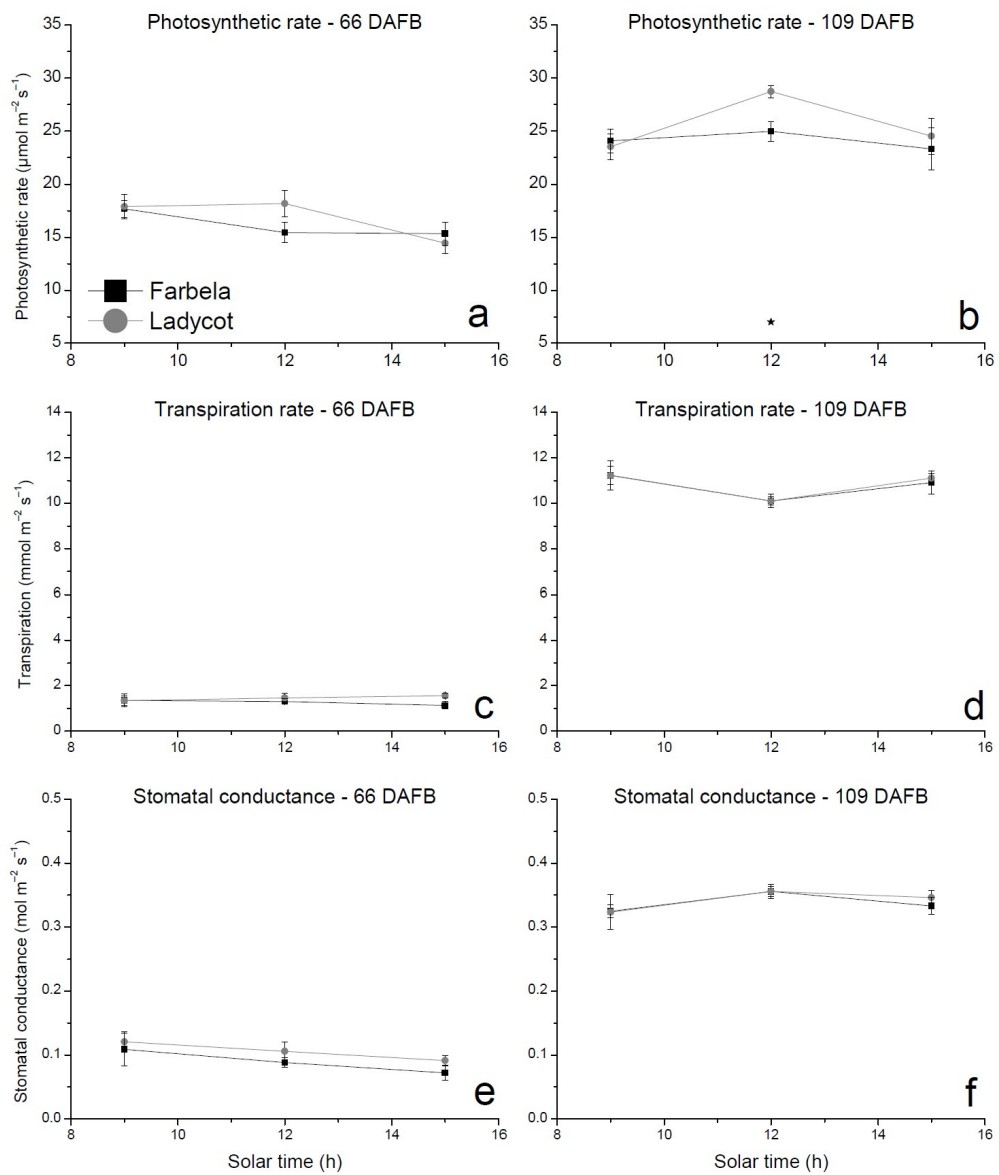

**Figure 6.** Daily patterns of leaf net photosynthesis (+SE), leaf transpiration (+SE) and stomatal conductance ($g_s$) (+SE) at 66 DAFB (**a,c,e**) and at 109 DAFB (**b,d,f**) for Farbela (squares) and Ladycot (circles). Each point represents the mean of 4–6 replicates. SE bars smaller than symbols not visible. Stars represent statistical differences for Student's *t* test $p < 0.05$.

### 3.4. Fruit Surface Conductance and Dry Matter

Fruit surface conductance, in general, decreased throughout the season even though the trend was reversed two weeks before harvest, showing an intense increase in both cultivars (Figure 7a). Ladycot perfectly followed this pattern, reaching its minimum of 1.8 m h$^{-1}$ at 95 DAFB and then showing a sharp increase up to 3.5 m h$^{-1}$ at harvest (Figure 7a). Farbela surface conductance showed a sinusoidal pattern: at 57 DAFB, it was 3.3 m h$^{-1}$, then decreased to 2.0 m h$^{-1}$ and again increased and decreased, with a minimum value of 1.0 m h$^{-1}$ at 115 DAFB. From this point on, two weeks before harvest, Farbela increased its surface conductance up to 2.1 m h$^{-1}$ at harvest (Figure 7a).

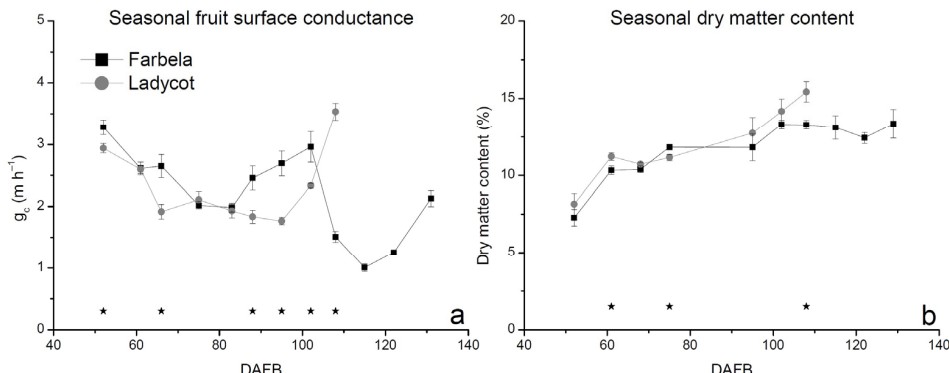

**Figure 7.** Seasonal fruit surface conductance (+SE) (**a**) and seasonal dry matter content (+SE) (**b**) for Farbela (squares) and Ladycot (circles). Each point represents the mean of 10 replicates. SE bars smaller than symbols not visible. Stars represent statistical differences for Student's *t*-test $p < 0.05$.

Dry matter content increased throughout the season for both cultivars (Figure 7b), reaching 15% in Ladycot, while Farbela reached 13% around 102 DAFB and maintained that value until harvest (Figure 7b). In the last four weeks before harvest, Farbela maintained a stable dry matter content (Figure 7b).

### 3.5. Sap Flow

Sap flow showed similar daily patterns during the season. It started to rise in the early morning, reaching high values at 9:00 h and continued to increase until mid-day (Figure 8b,d,f,h). In general, the sap flow daily patterns followed those of VPD (Figure 8b,d,f,h). At 59 DAFB daily sap flow reached a maximum value of 34.4 g m$^{-2}$ h$^{-1}$, while at 81 DAFB, it increased its maximum showing a value of 51.7 g m$^{-2}$ h$^{-1}$ at 13:00 h. In the mid-late stage, at 111 DAFB, a peak of 103.8 g m$^{-2}$ h$^{-1}$ was reached at 11:00 h (Figure 8b,d,f). At the end of the season, three days before harvest, sap flow showed another high peak at 14:00 h, although, as in the previous dates, the mean daily sap flow was significantly lower than the peak reached (Figure 8f,h).

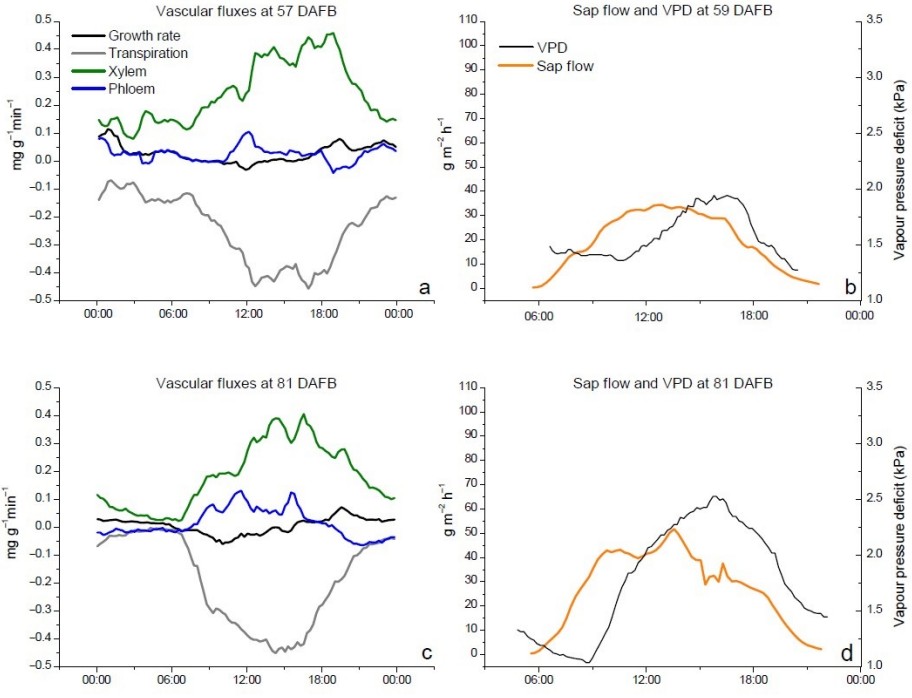

**Figure 8.** *Cont.*

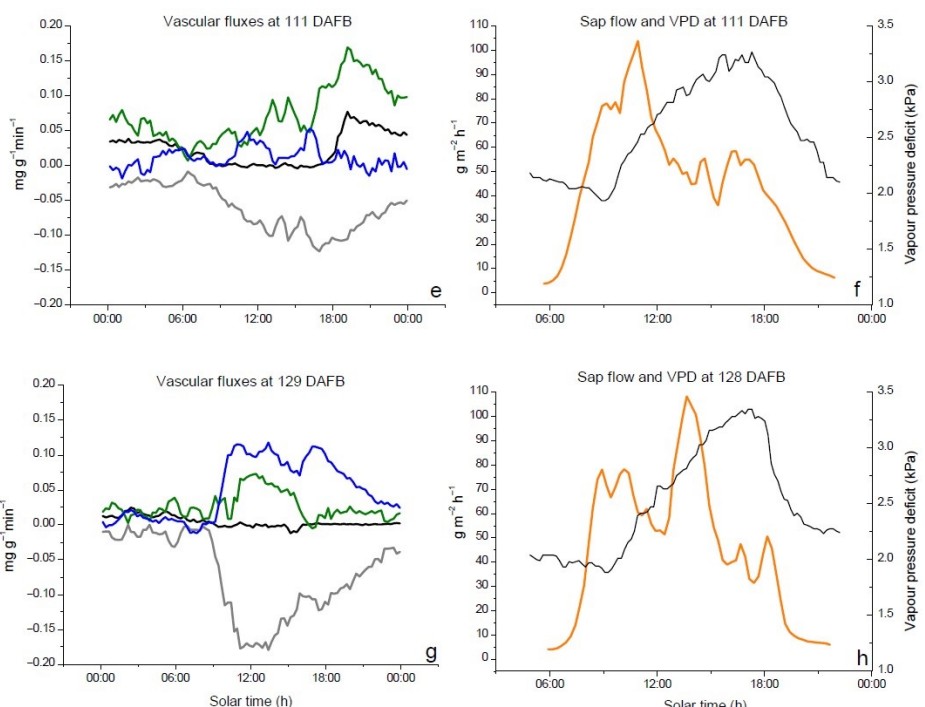

**Figure 8.** Average diurnal courses of fruit relative growth rate (RGR) and specific phloem, xylem, transpiration flow rates (mg g$^{-1}$min$^{-1}$) at 57 (**a**), 81 (**c**), 111 (**e**), and 129 (**g**) DAFB. Diurnal courses of VPD (kPa) and sap flow rate (g m$^{-2}$ h$^{-1}$) at 59 (**b**), 81 (**d**), 111 (**f**), and 128 (**h**) DAFB. For each parameter, data are the means of 4 replicates. Maximum SEs for transpiration were 0.122, 0.114, 0.028, and 0.068 at 57, 81, 111, and 129 DAFB, respectively. Maximum SEs for xylem were 0.13, 0.15, 0.03, and 0.06. Maximum SEs for phloem were 0.09, 0.07, 0.04, and 0.05. Maximum SEs for sap flow were 10.76, 20.49, 35.05, and 74.71.

*3.6. Seasonal Vascular and Transpiration Flows to/from the Fruit*

At 57 DAFB, fruit transpired 0.4 g water g$^{-1}$ d$^{-1}$ and xylem was the main inflow, allowing fruit growth despite the huge amount of transpiration water loss. The relative contribution of xylem to fruit growth was 91% of the total daily inflow, while fruit absolute growth was 1.2 g fruit$^{-1}$ d$^{-1}$ (Figure 9).

In the mid-stages (81 DAFB), fruit growth showed its minimum values in absolute terms. Fruit transpiration maintained high levels, around 0.3 g water g$^{-1}$ d$^{-1}$, corresponding to 9.2 g fruit$^{-1}$ d$^{-1}$, while both xylem and phloem flows decreased their daily specific and absolute import (Figure 9). At this stage, xylem and phloem accounted for about 92% and 8% of the fruit's total inflows, respectively, while 97% was lost by transpiration.

In the late stage (111 DAFB), during cell expansion, fruit growth increased while transpiration decreased to 0.1 g water g$^{-1}$ d$^{-1}$, corresponding to 6.6 g water fruit$^{-1}$ d$^{-1}$. Xylem contribution decreased from 0.3 to 0.1 g g$^{-1}$ d$^{-1}$ (7.694 g fruit$^{-1}$ d$^{-1}$), while phloem inflow increased its values (Figure 9). At this stage, phloem and xylem accounted for about 36% and 64% of the fruit's total inflows, while 55% was lost by transpiration.

In the last few days before harvest (129 DAFB), the fruit decreased its growth rate to 0.6 g fruit$^{-1}$ d$^{-1}$. Transpiration decreased but never reached values close to zero (0.10 g water g$^{-1}$ d$^{-1}$, which correspond to around 9.9 g water fruit$^{-1}$ d$^{-1}$). Xylem flows decreased, reaching their minimum values, while phloem flow increased its importance, showing values of 0.1 g g$^{-1}$ d$^{-1}$, corresponding to 6.9 g fruit$^{-1}$ d$^{-1}$ (Figure 9). At this stage, phloem and xylem accounted for about 66% and 34% of the fruit's total inflows, while 94% was lost by transpiration.

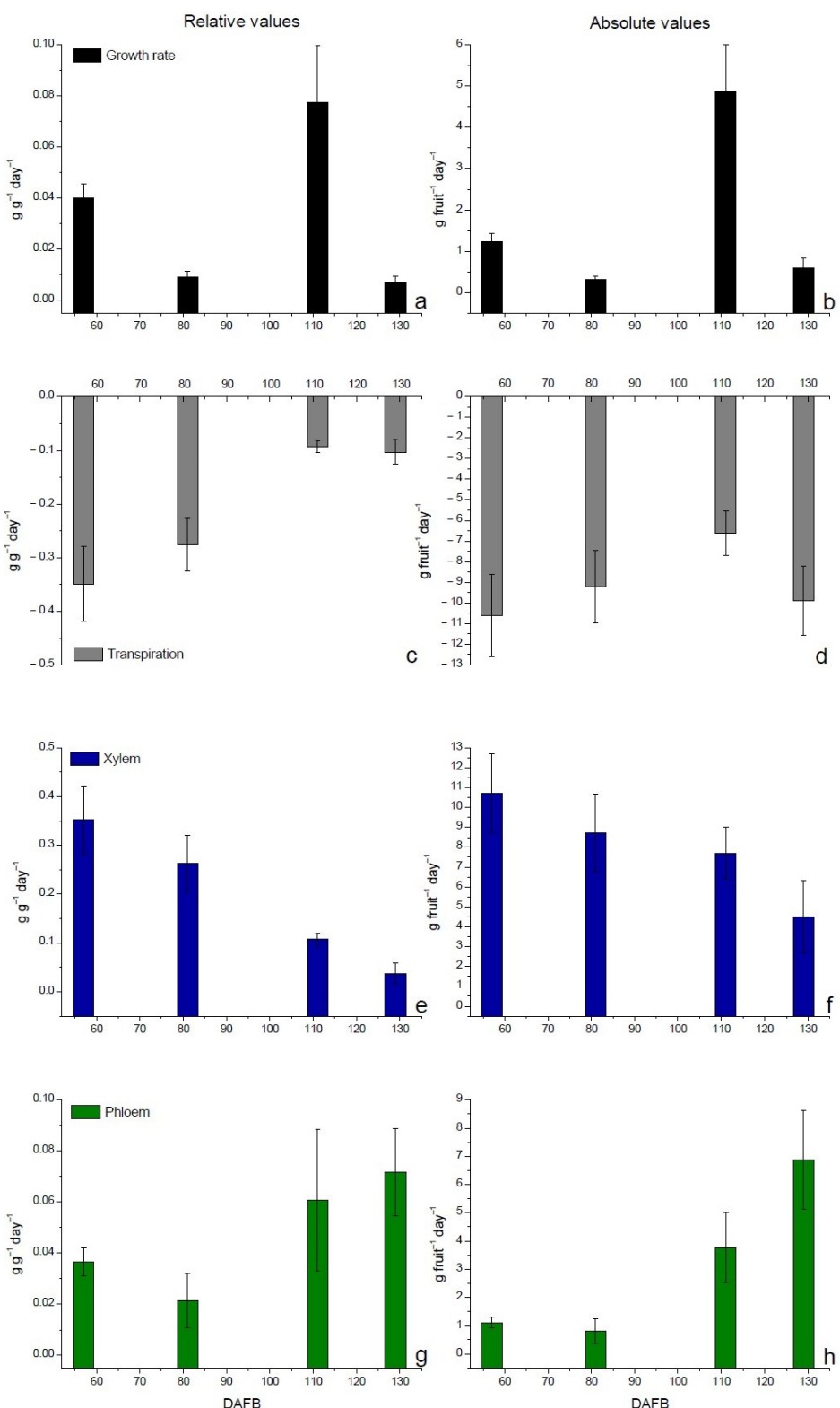

**Figure 9.** Mean (+SE) of the relative (RGR) (**a**) and absolute (AGR) (**b**) growth rates, specific (**c**) and whole fruit (**d**) transpiration, specific (**e**) and whole fruit (**f**) xylem flow, specific (**g**) and whole fruit (**h**) phloem flow at 57, 81, 111, and 129 DAFB.

### 3.7. Daily Vascular and Transpiration Flows to/from the Fruit

At 57 DAFB (early stage), young apricots grew at the highest rates during the late afternoon and the night, reaching a maximum of 0.1 mg $g^{-1}$ $min^{-1}$ after 18:00 h. Immedi-

ately after dawn, the fruit growth rate progressively slowed down to zero and maintained that value until 17:00 h. Around 12:00 h, the fruit showed a slight shrinkage (Figure 8a). Transpiration increased after dawn and stayed around 0.4 mg $g^{-1}$ $min^{-1}$ from 12:00 h to 17:00 h. In the same period, VPD reached its maximum levels (Figure 8a,b). The daily pattern of xylem flow mirrored transpiration; it increased after dawn, only managing to balance the transpiration losses. Thereafter, it decreased until the next morning (Figure 8a). Phloem flow maintained low values with several fluctuations during the day, showing inflows mostly around 12:00 h and during the night (Figure 8a).

At 81 DAFB (mid-stage), as in the earlier stage, fruit grew from late afternoon around 17:00 h to dawn, with a maximum RGR of 0.1 mg $g^{-1}$ $min^{-1}$. At this stage, the fruit shrank a little during the morning until 14:00 h (Figure 8c). Specific transpiration, on a daily scale, increased from dawn to late afternoon and showed maximum values of 0.5 mg $g^{-1}$ $min^{-1}$ at 14:30 h (Figure 8c); at the same time, VPD reached its maximum too (Figure 8d). Xylem flow mirrored transpiration but with lower values, not allowing to balance the transpiratory losses. After dawn, it increased and maintained its flow levels until evening (Figure 8c). Phloem inflow contributed to balance transpiration losses mostly in the central part of the day, reaching values up to 0.1 mg $g^{-1}$ $min^{-1}$ (Figure 8c).

At 111 DAFB (mid-late stage), fruit grew from late afternoon to dawn, while from dawn to afternoon, the balance between inflows and outflows was zero, so the fruit did not grow (Figure 8e). Specific transpiration rate was lower than those at 57 and 81 DAFB, with a maximum value of 0.1 mg $g^{-1}$ $min^{-1}$. Similarly to transpiration, xylem specific flow showed lower values than in the previous period, reaching a peak in the late afternoon (Figure 8e). Phloem contribution to fruit growth showed several fluctuations during the day, reaching maxima of 0.1 mg $g^{-1}$ $min^{-1}$ at 16:00 h. (Figure 8e).

At 129 DAFB (preharvest), fruit daily RGR showed slightly positive values during the night, whereas fruit shrank all day, with low daily net growth (Figure 8g). Specific transpiration rate reached 0.2 mg $g^{-1}$ $min^{-1}$ in the mid-day hours, while VPD was over 3 kPa (Figure 8h). Both xylem and phloem flows were not enough to sustain fruit growth during the light period; however, they almost recovered overnight (Figure 8g). At 129 DAFB, the fruit showed reduced growth but never reached zero (Figure 9a). Phloem flow maintained high values from 9:00 h to 21:00 h, while specific xylem flow decreased further.

## 4. Discussion

Apricot seasonal fruit growth showed the same double sigmoid pattern as other drupaceous species [5,38], with a pit hardening stage influenced by the length of the reproductive cycle. In fact, Ladycot showed a shorter stone hardening period compared with Farbela. In both cultivars, shoots alternated periods of fast and slow growth, stopping during the last part of fruit ripening. $\Psi_{stem}$ and $\Psi_{leaf}$ maintained values well above the stress threshold both in Ladycot and Farbela, even though towards the end of the season, values were lower than at the beginning. Leaves were able to sustain transpiration at mid-day, closing the stomata only in the afternoon. The higher levels of photosynthesis registered in the last part of the season could be the reason for the higher fruit growth rates, probably because of the higher partitioning toward the fruit instead of the shoots. In both cultivars, fruit epidermis remained permeable to water vapour during the whole season; in fact, the values of surface conductance were much above zero even close to harvest, as occurs in peach [15]. Apricot maintained high surface conductance, while many other species exhibit a progressive decrease, such as apple [13], grapevine [39], kiwifruit [40], Japanese plum [41], and sweet cherry [24,42]. These species are also characterised by higher decreases in xylem contributions than apricot. This may be due to both a decrease in the stem-to-fruit water potential gradient (as reported for grape [22,23]) and an increase in the hydraulic resistance of the xylem (e.g., anatomical dysfunctions in the xylem vessels reported in apple [43] and kiwifruit [44]). However, despite the decrease in transpiration throughout the season, apricot fruit xylem flow still maintained higher values (0.04 g $g^{-1}$ $d^{-1}$ or 4.50 g $fruit^{-1}$ $d^{-1}$) compared with apple and kiwifruit, for which almost null values are reported [21,40].

Apricot dry matter percentage showed variability based on the cultivar considered, as also found by Carbone et al. [45], who reported dry matter apricot values around 16% for "Orange Rubis" and 12% for "Bora". Ladycot reached 15% of dry matter content while Farbela had 13%, meaning that the first one was more efficient in the resource allocation process, but should also be considered the probable dilution effect that could have happened in Farbela towards harvest.

Sap flow and fruit transpiration followed the same pattern on a daily scale, although time shifts occurred because of physiological processes such as the rapid dehydration or rehydration of plant tissues [46]. Fruit transpiration reached its maximum values in the afternoon, in accordance with VPD. Based on our data, leaves received higher sap flow in the morning and at mid-day, while a higher amount of xylem water was moved to the fruit in the late afternoon. On the other hand, the fruit dry matter percentage remained stable in the last weeks before harvest despite the amount of phloem flows that reached the fruit, probably due to high fruit xylem inflows registered in the late season.

At the beginning of the season, xylem relative contribution to fruit growth was 91%, then it slightly increased up to 92% at 81 DAFB and finally slowly decreased throughout fruit cell expansion and ripening. In the late season, phloem increased its relative importance accounting for 36% and 66% of the daily relative contribution to fruit growth, at 111 and at 129, respectively. At stage I, apricot showed higher values in specific transpiration and xylem flow than those described by Morandi et al. [5] for peaches at the same stage, but since the phloem inflow was lower, net fruit growth was about half of that recorded for peach. Flows at this time of the season were similar to those described for young kiwifruit [18].

During the pit hardening stage, the xylem's relative importance to fruit growth was still higher than the phloem one, thanks to fruit transpiration and high surface conductance, as happens in peach [5]. Vascular flows and transpiration followed VPD during the day; higher inflows and outflows to and from the fruit were recorded when VPD started to increase. In the morning, after dawn, leaf water potential decreased, so leaves probably became a "stronger sink" for water than fruit; therefore, the fruit did not grow or even shrunk. When VPD increased in the afternoon, plants partially closed their stomata, so the stem-to-leaf water potential decreased during the late afternoon, while photosynthesis showed a decrease but remained high. Larger amount of xylem flow went to the fruit during the afternoon and continued during the evening and night to rehydrate the fruit and allow its growth. This daily pattern of xylem flow is similar to other species such as peach, kiwifruit, pear, and sweet cherry [5,17,18,24]. No cases of xylem backflow were recorded for apricot, probably because of the relatively high fruit transpiration rates, which helped lower fruit water potentials.

Fruit phloem inflows occurred mainly in the central part of the day, with peaks around midday, while low flows were observed during the night. Fruit phloem inflows did not respond to water potential gradients in the xylem but rather to the hydrostatic pressure gradient along the phloem pathway [8,10,11]. Decreased fruit hydrostatic pressure due to high water losses by transpiration in the midday hours could have possibly favoured a symplasmic phloem unloading at this time of the day. Symplasmic phloem unloading requires plasmodesmata to connect the phloem to fruit symplasts [11]. Recently Iqbal et al. [31] found apricot's main genes for specific sugar transporters. These genes are involved in sugar accumulation and transport to the fruit; therefore, this suggests the additional presence of an apoplasmic phloem unloading. In the last part of fruit development, a greater expression of sugar transport genes has been found [31], while at this time of the season, our data showed an increase in the importance of phloem flow to fruit growth. Therefore, the apoplasmic phloem unloading seems to be prevalent in the last part of the season when phloem flow becomes more important than in the early- and mid-stages. Passive and active mechanisms of phloem unloading may, in fact, coexist, as it has been hypothesised for other fruit species [47].

## 5. Conclusions

In this study, the seasonal and daily patterns of fruit growth, leaf gas exchanges, water relations, sap flows, and vascular flows were simultaneously monitored in apricot.

Results show that:

(1) In Farbela, leaves and fruit tended to import water at different times during the day. Leaves were "stronger sinks" for water during the morning. However, the fruit became a "stronger sink" during the afternoon, thanks to their high transpiration, which likely reduced fruit turgor pressure and allowed higher xylem flow towards them. Furthermore, midday phloem inflow peaks to the fruit could have led to a further increase in xylem flow in the afternoon because of an accumulation of osmotically active solutes. As a result, fruit growth occurred mainly during the late afternoon and night;

(2) In the early and mid-stages, the fruit growth strategy of Farbela was based on high water flows to and from the fruit. Transpiration and xylem flow decreased from the mid-stages but maintained relatively high values until harvest. In contrast, phloem flow became more important in the very last stages. Vascular flow showed the possibility of both symplasmic and apoplasmic phloem unloading; high water losses by transpiration could favour passive phloem unloading, while the recent report of several sugar transport genes in apricot suggests the simultaneous presence of an active phloem unloading as well.

(3) Despite the sap flow and fruit growth dynamics being monitored only in Farbela, Ladycot showed similar water relations and leaf gas exchanges throughout the season, despite differences in the shoot length. Moreover, the relatively high values of Ladycot fruit surface conductance led to the hypothesise that vascular flow dynamics in this cultivar might not be too different from those of Farbela.

This knowledge of the seasonal and daily behaviour of fruit vascular and transpiration flows could be an important starting point to improve apricot orchard fertilisation. Moreover, the knowledge of the different physiological behaviour in different growth stages will guide growers toward more precise and sustainable irrigation practices, allowing water savings without compromising fruit quality.

**Author Contributions:** Conceptualization, B.M., L.M. and L.C.-G.; methodology, B.M., L.M. and L.C.-G.; investigation, M.V., A.G. and S.G.-G.; data curation, M.V., A.G. and S.G.-G.; writing—original draft preparation, A.G.; writing—review and editing, M.V., S.G.-G. and B.M.; supervision, B.M. All authors have read and agreed to the published version of the manuscript.

**Funding:** This research received no external funding.

**Institutional Review Board Statement:** Not applicable.

**Informed Consent Statement:** Not applicable.

**Data Availability Statement:** The data presented in this study are available on request from the corresponding author.

**Conflicts of Interest:** The authors declare no conflict of interest.

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
