# Peer review of "Vascular and Transpiration Flows Affecting Apricot (Prunus armeniaca L.) Fruit Growth"

_agronomy, doi:10.3390/agronomy12050989_

Round 1

Reviewer 1 Report

Review of paper

Vascular and transpiration flows affecting apricot (Prunus armeniaca L.) fruit growth

By

Giovannini et al.

The paper reports a thorough investigation of fruit growth and fruit water relations in apricot. It is a well designed and well carried out study, the results reported are consistent and the conclusions drawn backed by the data.

I only have a few points that I suggest to the authors for considering when preparing a revised version. None of these points are critical.

  1. L 67 to 81: You could add the reference to sweet cherry [46] here.
  2. L 132 to 143: Please specify the day time when the measurements at the other DAFB were carried out - this is specified only for 66 and 109 DAFB.
  3. 208 and ff.
    1. I would be interested in knowing how these conductances compare to those published using Ficks law (P or g = F/(A*deltaC). This is a very common way researchers working on cuticle transpiration (Schreiber et al., Schönherr et al, Kerstiens et al. etc.) calculate cuticle permeances – the units are m s-1 which is identical to the units that you are using. The difference between your equation and those of the other group of scientists is just the term for the driving force. It may be interesting for the reader to be able to relate apricot skin conductance to published data on permeances of cuticles.
    2. I am wondering whether boundary layers are a factor in your conductance measurements. Did you use a fan for stirring? If not, your conductance estimates probably might represent those of the apricot skin plus a boundary layer. It would be nice if the reader had an idea on what the magnitude of such a layer would be.
    3. Not everybody is familiar with the term scalene ellipsoid – may be you indicate that this is an ellipsoid with three axes of different length.
  4. L 286 to 297, L. 300 to 307: I suggest to round to one digit. Please do so also in the subsequent text.
  5. 384: What about sweet cherries? See Planta 2001, 213, 927-936. https://doi.org/10.1007/s004250100568
  6. 381: What conductance are you referring to – O2, CO2 or water vapor? These are very different.
  7. Please check your contribution statement. X.X. Z.Z etc. – I assume this is an error.

This is a sound piece of work and I recommend this for publication in Agronomy.

Author Response

Review of paper

Vascular and transpiration flows affecting apricot (Prunus armeniaca L.) fruit growth

By

Giovannini et al.

The paper reports a thorough investigation of fruit growth and fruit water relations in apricot. It is a well designed and well carried out study, the results reported are consistent and the conclusions drawn backed by the data.

I only have a few points that I suggest to the authors for considering when preparing a revised version. None of these points are critical.

  1. L 67 to 81: You could add the reference to sweet cherry [46] here. Thank you for the suggestion, we have added a short paragraph describing sweet cherry fruit vascular fluxes. L 70 to 73.
  2. L 132 to 143: Please specify the day time when the measurements at the other DAFB were carried out - this is specified only for 66 and 109 DAFB. We have changed the sentence, underling that the measurements were taken at solar midday. L138.
  3. 208 and ff.
    1. I would be interested in knowing how these conductances compare to those published using Ficks law (P or g = F/(A*deltaC). This is a very common way researchers working on cuticle transpiration (Schreiber et al., Schönherr et al, Kerstiens et al. etc.) calculate cuticle permeances – the units are m s-1 which is identical to the units that you are using. The difference between your equation and those of the other group of scientists is just the term for the driving force. It may be interesting for the reader to be able to relate apricot skin conductance to published data on permeances of cuticles. The values showed in this article should be comparable to those of already pubblished data, with of course the need to convert m h-1 (used in this article) to m s-1. However, we had not sufficient time to re-calculate our values. For this reason we do not feel confident to mention this possible comparison in the text.
    2. I am wondering whether boundary layers are a factor in your conductance measurements. Did you use a fan for stirring? If not, your conductance estimates probably might represent those of the apricot skin plus a boundary layer. It would be nice if the reader had an idea on what the magnitude of such a layer would be. We did not use a fan for stirring, since in our area wind is not common weather condition. Hence, we performed the fruit surface conductance measurements maintaining the fruit in the most similar conditions to those experienced in the field.
    3. Not everybody is familiar with the term scalene ellipsoid – may be you indicate that this is an ellipsoid with three axes of different length. Yes, we corrected it. L224-225.
  4. L 286 to 297, L. 300 to 307: I suggest to round to one digit. Please do so also in the subsequent text. Yes, we have rounded all the values to one digit.
  5. 384: What about sweet cherries? See Planta 2001, 213, 927-936. https://doi.org/10.1007/s004250100568. We have added a mention also to sweet cherry. L.436.
  6. 381: What conductance are you referring to – O2, CO2 or water vapor? These are very different. We have underlied that the fruit remained permeable to water vapour since it was able to sustain high transpiration rates even close to harvest. L. 432.
  7. Please check your contribution statement. X.X. Z.Z etc. – I assume this is an error. Thank you for the comment, we have checked the contribution statement, also adding some information.

This is a sound piece of work and I recommend this for publication in Agronomy.

Reviewer 2 Report

The current stady entitled „Vascular and transpiration flows affecting apricot (Prunus armeniaca L)” is very important in order to explain the growth strategy of apricot fruits on a seasonal and daily scale, determine their water relations, leaf gas exchange and determine quantifying xylem, phloem and transpiration to and from the fruit, which, despite the current state of knowledge on this subject, is not fully known. This knowledge can be of great practical importance in improving the management of apricot orchards in terms of their fertilization and irrigation.

The article presented to me for review contains all the key elements required in the scientific work and editorial work of the journal, but I have some minor suggestions for the improvement of the manuscript:

Abstract

  • Suggestion to provide the future prospects and potential benefits of this study in one line at the end of the abstract.

Conclusions

  • Conclusions should include the potential benefits of choosing this solution, as well as recommendations and prospects for the future. Meanwhile, at some points they contain elements of discussions.

References

  • The reference number 43 in line 580 is nowhere quoted in the text.

Author Response

The current study entitled „Vascular and transpiration flows affecting apricot (Prunus armeniaca L)” is very important in order to explain the growth strategy of apricot fruits on a seasonal and daily scale, determine their water relations, leaf gas exchange and determine quantifying xylem, phloem and transpiration to and from the fruit, which, despite the current state of knowledge on this subject, is not fully known. This knowledge can be of great practical importance in improving the management of apricot orchards in terms of their fertilization and irrigation.

The article presented to me for review contains all the key elements required in the scientific work and editorial work of the journal, but I have some minor suggestions for the improvement of the manuscript:

Abstract

  • Suggestion to provide the future prospects and potential benefits of this study in one line at the end of the abstract. Yes, we have added a short sentence about the benefits and prospects of the study. L 29-31.

Conclusions

  • Conclusions should include the potential benefits of choosing this solution, as well as recommendations and prospects for the future. Meanwhile, at some points they contain elements of discussions. We have modified the text trying to eliminate some discussion point while summarizing the main findings. Moreover, we have expanded the section about potential benefits and future prospects. L 526-528.

References

  • The reference number 43 in line 580 is nowhere quoted in the text. We have revised the text and deleted that citation.

Reviewer 3 Report

Please check the resolution of the graphs

  • the figures (Fig.1 to Fig.9) have a very low resolution, please check it, maybe is a problem only with the proof.

Author Response

Please check the resolution of the graphs the figures (Fig.1 to Fig.9) have a very low resolution, please check it, maybe is a problem only with the proof. Thank you for the comment. I will get in touch with the editorial office to know whether the original files I submitted have sufficient quality or if I need to improve them.